# Persistence of phenotypic responses to short-term heat stress in the tabletop coral *Acropora hyacinthus*

Nia S. Walker[1]*, Brendan H. Cornwell[1], Victor Nestor[2], Katrina C. Armstrong[1], Yimnang Golbuu[2], Stephen R. Palumbi[1]

1 Department of Biology, Hopkins Marine Station of Stanford University, Pacific Grove, California, United States of America, 2 Palau International Coral Reef Center, Koror, Palau

* niawalker13@gmail.com

**Data Availability Statement:** All relevant data are within the paper and its Supporting Information files.

## Abstract

Widespread mapping of coral thermal resilience is essential for developing effective management strategies and requires replicable and rapid multi-location assays of heat resistance and recovery. One- or two-day short-term heat stress experiments have been previously employed to assess heat resistance, followed by single assays of bleaching condition. We tested the reliability of short-term heat stress resistance, and linked resistance and recovery assays, by monitoring the phenotypic response of fragments from 101 *Acropora hyacinthus* colonies located in Palau (Micronesia) to short-term heat stress. Following short-term heat stress, bleaching and mortality were recorded after 16 hours, daily for seven days, and after one and two months of recovery. To follow corals over time, we utilized a qualitative, non-destructive visual bleaching score metric that correlated with standard symbiont retention assays. The bleaching state of coral fragments 16 hours post-heat stress was highly indicative of their state over the next 7 days, suggesting that symbiont population sizes within corals may quickly stabilize post-heat stress. Bleaching 16 hours post-heat stress predicted likelihood of mortality over the subsequent 3–5 days, after which there was little additional mortality. Together, bleaching and mortality suggested that rapid assays of the phenotypic response following short-term heat stress were good metrics of the total heat treatment effect. Additionally, our data confirm geographic patterns of intraspecific variation in Palau and show that bleaching severity among colonies was highly correlated with mortality over the first week post-stress. We found high survival (98%) and visible recovery (100%) two months after heat stress among coral fragments that survived the first week post-stress. These findings help simplify rapid, widespread surveys of heat sensitivity in *Acropora hyacinthus* by showing that standardized short-term experiments can be confidently assayed after 16 hours, and that bleaching sensitivity may be linked to subsequent survival using experimental assessments.

**Funding:** This study was supported by National Science Foundation NSF grant OCE [1736736]; SRP and YG received funding. https://www.nsf.gov/ The funders had no role in study design, data collection and analysis, decision to publish, or preparation of the manuscript.

**Competing interests:** The authors have declared that no competing interests exist.

## Introduction

There is urgent need for research that aims to uncover the mechanisms leading to coral stress resilience [1, 2], which is the ability of these keystone organisms to survive variable and hostile environments. Whether corals are resilient in the face of mounting environmental challenges depends on several biotic and abiotic factors, including respective tolerance limits of corals and dinoflagellate endosymbionts and coral host-symbiont interactions [3–6], coral species identity [7, 8], nutrient availability [9], and the duration and intensity of stressors [10–12].

The importance of resilience to future reef function has led to calls for increased attention toward mapping heat resistance across and within species [13]. To accommodate the large number of experiments needed for such mapping, researchers have recently focused on experiments that impose short pulses of heat exposure mimicking high temperatures in shallow waters at noon time low tides [13–18]. These heat pulse experiments impose high heat for short periods and then typically assay coral bleaching the morning after [reviewed in 17]. Based on these experiments, immediate impacts of heat stress on corals can be observed in the transcriptome, including upregulation of genes associated with the immune response and apoptosis [19–22]. Symbiont impacts, for example, cellular structure damage and oxidative stress during and hours after exposure to stressors, may also influence coral holobiont survival [23–26].

Coral heat resistance studies are primarily concluded within the first 16 hours post-heat stress [27], and there is comparatively little information about heat stress impacts beyond the first 24 hours and over the first few days after a heating event. Better understanding heat stress impacts beyond the immediate effects and hours afterwards would further illuminate how the coral holobiont manages stress and then transitions from a stressed to a recovery state. This may provide further validation for the utility of such short-term experiments and rapid assays in the extensive reef mapping projects that may be necessary to find, protect, and manage future reefs. Extending the timeline of heat stress experiment observation could also result in directly linking coral heat stress resistance and recovery, which may be especially important when there is high variation in resistance and recovery ability within and between species.

In this study, we investigated early phenotypic responses of coral fragments to heat stress collected from individual colonies of *Acropora hyacinthus*—an abundant reef-building coral species with extensive geographic distribution that is representative of many widespread, bleaching sensitive taxa inhabiting tropical reefs [13, 28–30]. We employed a two-day short-term heat stress experiment followed by daily monitoring of bleaching intensity and mortality for seven days, to test the stability of the bleaching response and whether bleaching severity is linked to higher likelihood of mortality shortly after heat stress. We additionally revisited the coral fragments approximately one and two months after heat stress to determine how variation in heat stress resistance and short-term impacts of the heat stress response affected longer-term recovery.

To allow for repeated and non-destructive bleaching measurements, we employed a five-point visual bleaching score (VBS) system based on coloration relative to baseline coral fragment color throughout the post-heat stress period [13]. This qualitative bleaching method allowed for simple, undisturbed observation of sample bleaching state following heat stress. As with other studies that used visual bleaching metrics [31–36], we corroborated results with a quantitative metric counting symbiont cells in proportion to total holobiont cells. Therefore, our daily observational study was able to provide further insight into the heat stress response by examining this short timeframe following heat stress. This study highlights the importance of combined coral heat stress resistance and recovery assays and considering survival and recovery on short timescales.

## Materials and methods

### Coral colony sampling

We sampled 101 *Acropora hyacinthus* colonies located on 28 reef locations in Palau's northern and southern lagoons (S1 Fig). We selected on average 3–4 genets per reef that were at least 5 meters apart to limit co-collection of clonal colonies (unique genets supported by mitochondrial genome sequencing, described further in S1 Table), a subset of genets from a larger coral reef survey program [13]. Sampling took place from 23rd July to 10th August 2018 (S1 Table). The heat stress experiments were conducted on an ongoing basis, i.e., coral fragments were added to tanks as they were collected over staggered days. There were 15–25 fragments per heat stress tank, and fragments at different stages of the heat stress experiment (i.e., heat ramp day 1 vs. day 2) did not overlap in tanks. The primary intention of widespread sampling was to capture heat resistance variability among a set of previously sampled corals with known resistance history [13]. All necessary collection permits were obtained from the following authorities: Republic of Palau Ministry of Natural Resources, Environment, and Tourism (Marine Research Permit), Aimeliik State Government, Koror State Government (Rock Island Permit), Kayangel State Government, and Ngarchelong State Government. All samples were exported to the USA through CITES permit PW19-011, issued by the Republic of Palau Bureau of Marine Resources.

Temperature on each reef was recorded from 8th November 2017 to 8th August 2018 at 10 min intervals (HOBO, OnSet Computing, Massachusetts). Loggers were placed adjacent to a subset of genets on reefs. Temperature data were averaged for reefs with multiple retrieved loggers; replicate loggers were analogous within reefs. Loggers were also irretrievable from 8 out of 28 reef sites (S1 Table). To quantify temperature differences for analysis, we counted the number of events above 31˚C—this threshold allowed us to widely compare temperature spikes across reefs, and other thermal spike thresholds yielded similar results (e.g., 29˚C and 30˚C) [37]. There were no recorded mass bleaching events in Palau during this collection period, though there were mild levels of accumulated heat stress recorded in 2018 (NOAA Coral Reef Watch) [38]. Our data found that reef temperatures had little variability and did not exceed 32.5˚C [37].

We sampled fist sized fragments from genets, loosely wrapped them in bubble wrap that was previously soaked in seawater [39], and then transported samples by boat to the Palau International Coral Reef Center (PICRC). We then placed these coral fragments into large flow-through holding tanks and clipped them further into four ~5cm length ramets per genet. The following day, we scored all ramets for bleaching (visual bleaching score, described in the following section) and photographed them prior to beginning the short-term heat stress experiment.

### Metrics for bleaching severity and mortality

To measure corals repeatedly without destructive sampling we used a visual bleaching score (VBS) method based on a five-point scale: (1) no bleaching, (2) slightly discolored, with a small amount of visible bleaching, (3) moderately discolored, clearly bleaching, (4) severely discolored, nearly complete bleaching but with some remaining color, (5) no color, total bleaching (Fig 1) [see also 13]. The VBS metric was used before and after the short-term heat stress experiment, daily for one week following bleaching, and when we examined corals after approximately one and two months of recovery. We scored ramets using two observers for all assessments up to a week post-stress. Only one in-person observer was available for one and two months post-stress timepoints, and these scores were confirmed with photographs.

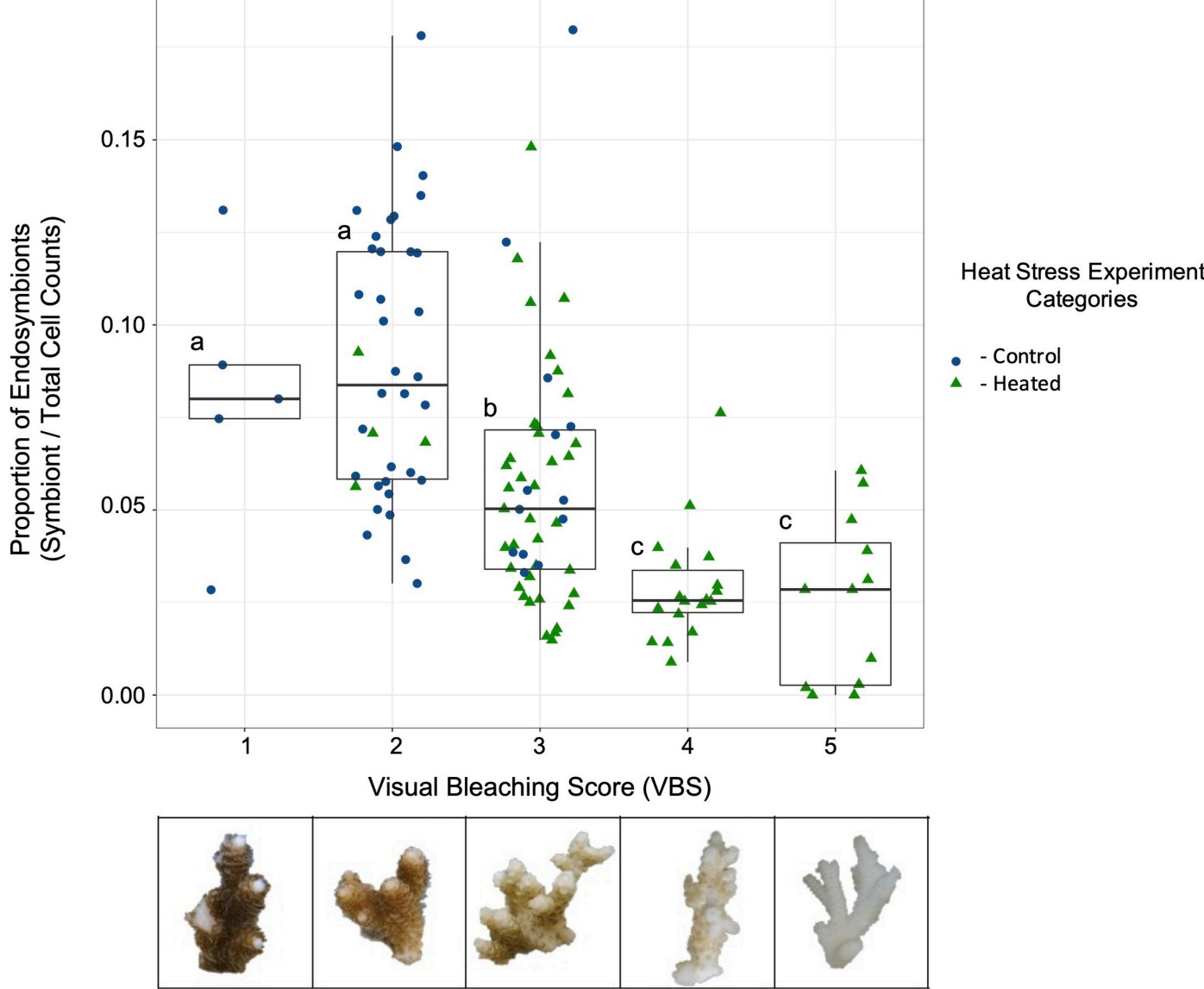

**Fig 1. Relationship between symbiont proportion and visual bleaching score metrics after the heat stress experiment.** Quantitative measure of bleaching (identified symbiont cells divided by total cell counts) versus qualitative visual bleaching scores (VBS) of sacrificed ramets. Below the x-axis is an example photo representation of the different bleaching severity categories. All sacrificed control (blue circles) and heat stressed (green triangles) samples were included to increase sample size across visual bleaching scores. An ANOVA and post-hoc Tukey tests were performed between all visual bleaching score categories, with reef region included as a random intercept to account for spatial variability. We found that categories segregated based on the three labeled groups—a, none to little bleaching; b, moderate bleaching; and c, severe to total bleaching ($p < 0.05$) (S2 Table).

Photographs were taken at the same approximate position and time of day, while monitoring bleaching and mortality. Mortality was determined by examining each ramet for any presence of tissue on the skeleton. We divided mortality into the following categories: not dead (i.e., none or some visible tissue absence) and dead (i.e., complete absence of tissue).

Using visual bleaching scores allows for rapid assessment of many corals over consecutive time points and is valuable for long-term assays. However, visual scoring may be subjective among observers or over time, potentially leading to variability or inaccuracy of results. To gauge the value of visual bleaching scores and quantify the proportion of symbiont cells out of

total holobiont cells after the heat stress experiment, we used flow cytometry (Guava EasyCyte HT; Millepore, Massachusetts) to count the proportion of symbiont cells to total cells in a ramet [13, 40]. Immediately after removal from the tanks we airbrushed coral tissue from the skeleton in seawater, centrifuged the slurry and resuspended it in RNAlater. After transport back to Hopkins Marine Station (USA), we washed the RNAlater-tissue suspension in DI water and then resuspended the tissue pellet in a 0.01% SDS:deionized water solution. Samples were homogenized with the PowerGen rotostat for 5 seconds at the highest setting, needle sheared to break apart cell clumps, then diluted by 1:200 in 0.01% SDS and run in triplicate on the flow cytometer. We firstly gated counts on the forward scatter channel (FSC) to exclude small particles (less than $10^2$ fluorescent units). Next, we gated all events that exceeded $10^4$ fluorescent units on the 690 nm detector as symbiont counts [40]. We subtracted events detected in the negative control (0.01% SDS), then we calculated the symbiont proportion as number of symbiont gated events divided by the total number of events after the first FSC gate. We were able to successfully quantify symbiont concentration in 164 fragments (n = 85 controls and n = 79 heat stressed samples).

## Short-term heat stress experiment

The short-term heat stress experiments ran in 10 L coolers outfitted with constant seawater inflow (ca. 1/2 volume h$^{-1}$) from the surrounding southern lagoon with large particles filtered out. Tanks were equipped with two chillers (Nova Tec, Maryland), a 300W heater, a submersible water pump (~280 L h$^{-1}$), and overhanging LED light fixtures (Apogee Instruments Underwater Quantum Flux meter model MQ-210, ca. 53–94 µmol photons m$^{-2}$s$^{-1}$) set to a 12 h light:dark cycle [methods described further in 13]. Starting 22nd July 2018, heated coral ramets ramped from 30˚C to 34.5˚C over three hours (1000 to 1300 hrs), held at 34.5˚C for three hours (1300 to 1600 hrs), then ramped down to 30˚C (1600 to 1800 hrs) and held at 30˚C until the next day (S2 Fig). Two experimental ramets were subjected to two days of this ramp cycle while two control ramets sat in separate identical coolers that remained at 30˚C for two days per genet. The morning after the two-day heat stress experiment, all experimental and control ramets were scored via visual bleaching score (VBS) and photographed. One replicate experimental and one control ramet per genet were sacrificed in RNAlater for cell counting via flow cytometry. See Cornwell et al., (2021) for evidence of low symbiont concentration variation between two replicate ramets from these individual colonies, though further sampling may improve characterization of colony heat resistance [13]. The remaining replicate experimental and control ramets were placed into a large holding tank for the post-stress experiment period.

## Post-heat stress experiment

One large outdoor flow-through 760 L holding tank with turnover of one full volume per hour was used for short- and long-term monitoring of recovery. Seawater inflow also came from the surrounding southern lagoon, and tank temperature was periodically recorded (~28.5–30˚C). The holding tank was kept underneath a large roof that provided some light protection, though no additional shade devices were installed to mitigate further light damage, nor were additional light fixtures included. We used water pumps for circulation and relied on natural sunlight, and we did not provide supplemental feeding to corals in the recovery holding tank. Ramets were either epoxied upright onto plastic crates or laid flat down on egg crate. There were no observable differences in survival between methodology among moderate (VBS 3) or high (VBS > 3) bleaching severity corals (Fisher's Exact Test, respectively, $p$ = 0.7342 and $p$ = 1, S2 Table). We did find a significant difference in survival between epoxy and egg crate

laid low bleaching severity corals (VBS < 3), though this likely resulted from the relatively low sample size of epoxied corals (8 compared to 14 egg crate laid samples) and low mortality (only 5 out of 22 samples) (Fisher's Exact Test, $p$ = 0.03934). Experimental and control ramets from each genet were kept next to each other in identical conditions. Ramets were added to the holding tank as they finished the heat stress experiment. The morning after the two-day heat ramp was called Day 0 of the post-stress period. At approximately 8:00AM daily until Day 7 post-stress, all samples were scored for their survival and bleaching severity (via VBS) and were photographed. In order to evaluate visual recovery and mortality approximately one- and two-months post-heat stress, we scored and photographed all corals on: 22nd August, 4th September, 7th September, 10th September, 13th September, and 2nd October. On 10th September 2018, all samples were moved into an adjacent, comparable flow-through holding tank to remove macroalgal buildup on the previous tank's walls. No macroalgae were removed from the samples or plastic crate they rested on.

## Statistical analyses

We ran all statistical analyses in R (version 4.0.5). We tested for differences between heat stressed and control samples' bleaching severity (Day 0 post-stress, n = 88 heat stressed and n = 101 controls out of 101 genets) and mortality (Day 7 post-stress, n = 101 heat stressed and n = 101 controls out of 101 genets) using Pearson's chi-square test. We used a linear mixed effects model to determine whether quantitative symbiont concentration values correlated with qualitative bleaching scores, and we compared symbiont concentration of visual bleaching score groups using a one-way ANOVA and post-hoc Tukey test. We used a linear mixed effects model to evaluate change in bleaching severity among heat stressed corals from Day 0 to Day 7 of the post-stress period (n = 50 genets). We also used a one-way ANOVA and post-hoc Tukey test to evaluate any changes in bleaching on Post-Stress Day 7, Month 1, and Month 2 among samples with different levels of heat resistance (based on Day 0 bleaching severity). Further, we ran a mixed effects logistic regression to predict whether bleaching severity on Day 0 influenced likelihood of mortality two months post-stress (n = 88 heat stressed genets). We collected corals from a wide geographic range across Palau's northern and southern lagoons to capture diverse heat stress responses. When evaluating heat resistance and mortality among samples, we accounted for any possible spatial variability between reef regions (see S1 Fig) by using linear mixed effects models (linear regressions and ANOVAs, R package nlme; logistic regression, R package lmer) that each specified reef region as a random intercept. Marginal $R^2$ (based on fixed effects) and conditional $R^2$ (based on fixed + random effects) values were calculated using R package sjstats. Post-hoc Tukey tests for linear mixed effects models were performed using R package multcomp. Lastly, we also used an ordinal logistic regression to evaluate whether reef temperature might have significantly influenced bleaching severity categories (low: n = 13, moderate: n = 28, high: n = 18 heat stressed fragments) and mortality (n = 8) during the heat stress experiment. All model formulas and outputs can be found in S2 Table.

## Results

### Ground truthing VBS with flow cytometry

Symbiont proportion quantitatively tracked visual bleaching scores, although there was high variation: low scores (none and visible) were highly distinct from moderate scores and from high scores (severe and total) (ANOVA and post-hoc Tukey test, all comparisons $p < 0.005$, Fig 1 and see full statistical results in S2 Table). In addition, there was little distinction between no bleaching (VBS 1) and visible bleaching (VBS 2) (ANOVA and post-hoc Tukey test,

$p = 0.94141$), and between severe (VBS 4) and total bleaching (VBS 5) when comparing symbiont proportion averages (ANOVA and post-hoc Tukey test, $p = 0.99708$). On average, corals with little bleaching (VBS 1 and 2) had 8.9 ± 3.6% symbiont proportion, moderately bleached corals (VBS 3) had 5.8 ± 3.4% symbionts, and severely bleached corals (VBS 4 and 5) had 2.8 ± 1.8% symbionts (Fig 1). These results suggest strong confidence in determining bleaching severity based on visual bleaching scores between minimal and severe bleaching. However, these results also reveal that smaller changes in visible bleaching (i.e., VBS 1 vs 2 and VBS 4 vs 5) as determined with the visual bleaching score method do not correlate well with symbiont proportion. It is worth noting that sample size may have played a role in the ability to accurately capture distinctions between VBS categories—the two categories with the fewest coral samples were those with no bleaching (VBS 1, 5 samples) or total bleaching (VBS 5, 12 samples) (S1 Table).

## Variable bleaching severity in corals immediately following short-term heat stress

After two days of short-term heat stress, 88 of 101 heated genets remained alive but showed a wide variety of bleaching results. Overall, we observed bleaching in 95% of heated genets: most genets showed visible (VBS < 3; 17%), moderate (VBS 3; 43%) or severe (VBS > 3; 25%) bleaching (S3 Fig) with one coral showing no bleaching and five being totally bleached. Due to the above quantitative flow cytometry results, corals were partitioned into three bleaching severity categories based on these visual bleaching scores: 22 low bleaching severity (none or visible bleaching), 38 moderate severity (moderate bleaching), and 28 high severity (severe or total bleaching). We excluded the 13 dead coral genets from all further bleaching severity analysis.

Controls were similarly evaluated to test for potential negative impacts of sample collection and transport to the laboratory. All controls survived while in the stress tank system, and 96 out of 101 controls showed no or minimal bleaching (S3 Fig). Analysis of controls was particularly important, because samples were collected over a period of 19 days and tested in staggered experiments due to the widespread geography of the reefs sampled. As a result, similar survival and lack of bleaching among controls suggested that experimental artifacts of sampling and handling did not significantly impact observed bleaching severity or mortality in heat stressed samples.

## Rapid bleaching results predict bleaching and mortality after one week

After following all bleached corals over the course of one week, we found that regardless of the heat resistance category (low, moderate, or high bleaching) surviving genets had highly stable bleaching severity over time (Fig 2). The majority of all samples (90%, n = 45 out of 50 genets) remained in the same bleaching severity category from Day 0 to Day 7 post-stress. Of the five genets that switched bleaching severity categories, four improved: 2 moderate → low bleaching and 2 high → moderate bleaching. The other genet worsened from moderate to high bleaching. As a result, we found a highly significant positive relationship between visual bleaching scores on Day 0 and Day 7 post-stress (Fig 3A and S2 Table, linear mixed effects model, $p < 2.2e^{-16}$, marginal $R^2 = 0.863$, df = 44), such that VBS at Day 0 post-stress was highly predictive of VBS throughout the following seven-day period.

Mortality post-stress was also predictable after 16 hours. By the end of one-week post-heat stress, 51 total genets had died (50.5% mortality), including 38 (43.2%) among the 88 genets that survived immediate impacts from the heat stress experiment. Further, more than 50% of total mortality occurred within two days of the heat stress experiments. Genets with little to no

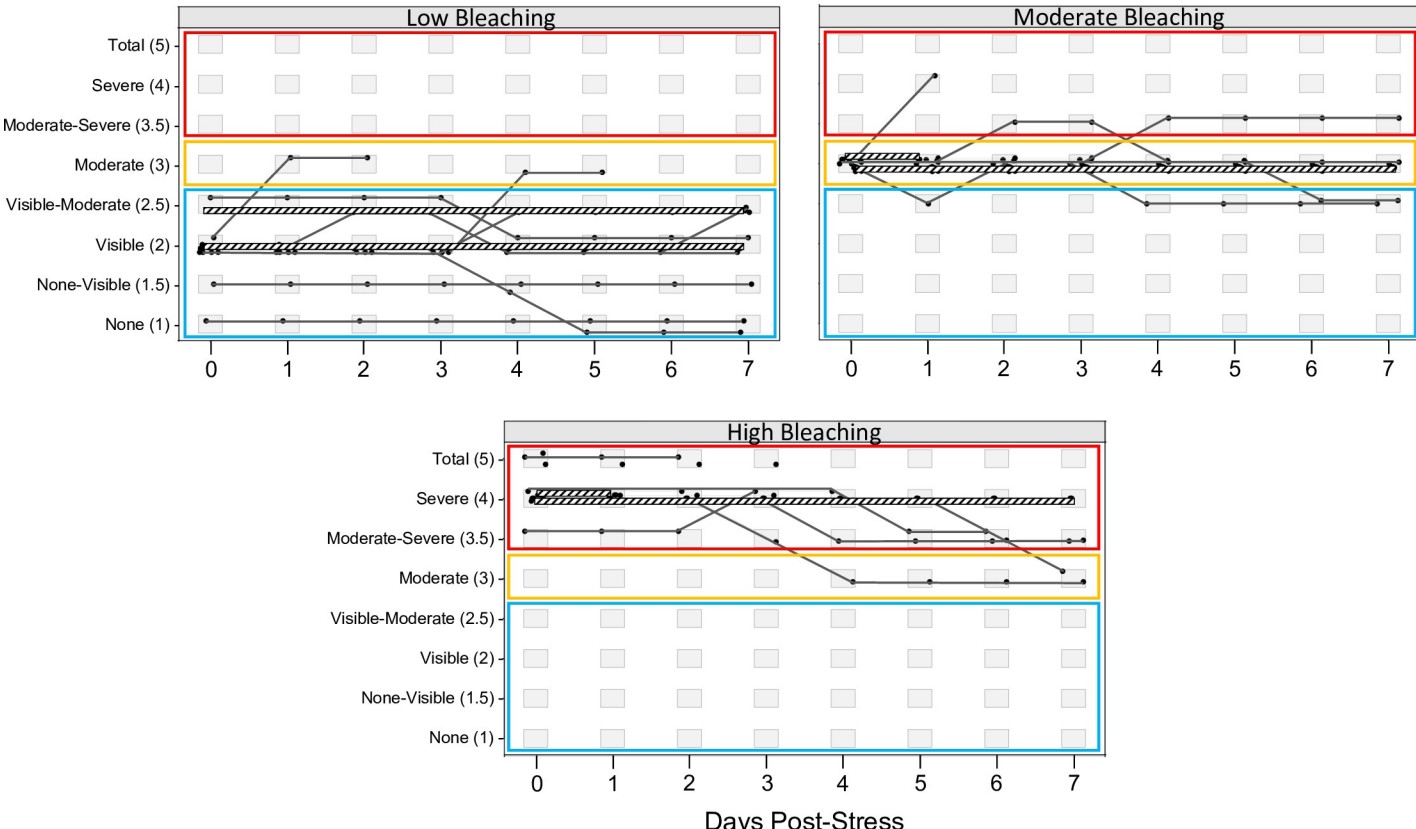

**Fig 2. Bleaching stability over one week following heat stress.** Ordinal time-series scatterplots, showing bleaching severity and visual bleaching scores of heat stressed corals daily from Days 0 to 7 of the post-stress period. From left to right, samples were divided into their Day 0 post-stress bleaching severity groups (low, moderate, and high). Each point on a given day represents one coral genet, and truncated lines represent mortality of those genets. Black cross-hatch lines highlight the 50% most frequent trajectories, whereas thin gray lines are low support trajectories. Each plot includes three boxes to show the bleaching severity groups: blue = low, yellow = moderate, and red = high. Plots were made using otsplot within the R package vcrpart.

bleaching on Day 0 survived best: only 4 out of 22 (18%) genets in this category died, three of them by Day 2. Genets showing moderate bleaching had higher mortality: 17 out of 38 genets died (45%), 12 of these by Day 2. The most heat susceptible genets experienced the highest mortality: 30 out of 41 (73%, including those that sustained severe to total bleaching or died during heat stress) (Fig 3B). Overall, bleaching severity on Day 0 post-stress was significantly and positively correlated with likelihood of mortality by Day 7 (S4A Fig, mixed effects logistic regression, $p = 0.00184$, pseudo marginal $R^2 = 0.165$) and Month 2 (Fig 3C, logistic regression, $p = 0.00414$, pseudo marginal $R^2 = 0.135$), although relatively low $R^2$ values show that much of the variation remained unexplained.

## Environmental differences in bleaching resistance

Previous work in Palau showed widespread occurrence of heat resistance, though presence of heat resistant corals in part coincided with thermal environment [13]. Consistent with Cornwell et al., (2021), we found that bleaching resistance was widespread throughout Palau's geographic regions [13]. There were minimally bleached genets that originated from 11 patch reefs (7 in the south and 4 in the north) and three fore reef locations (1 in the south and 2 in the north). Four of these patch reefs (2 in the south and 2 in the north) had all categories: low, moderate, and high bleaching severity. Genets that sustained exclusively moderate or worse

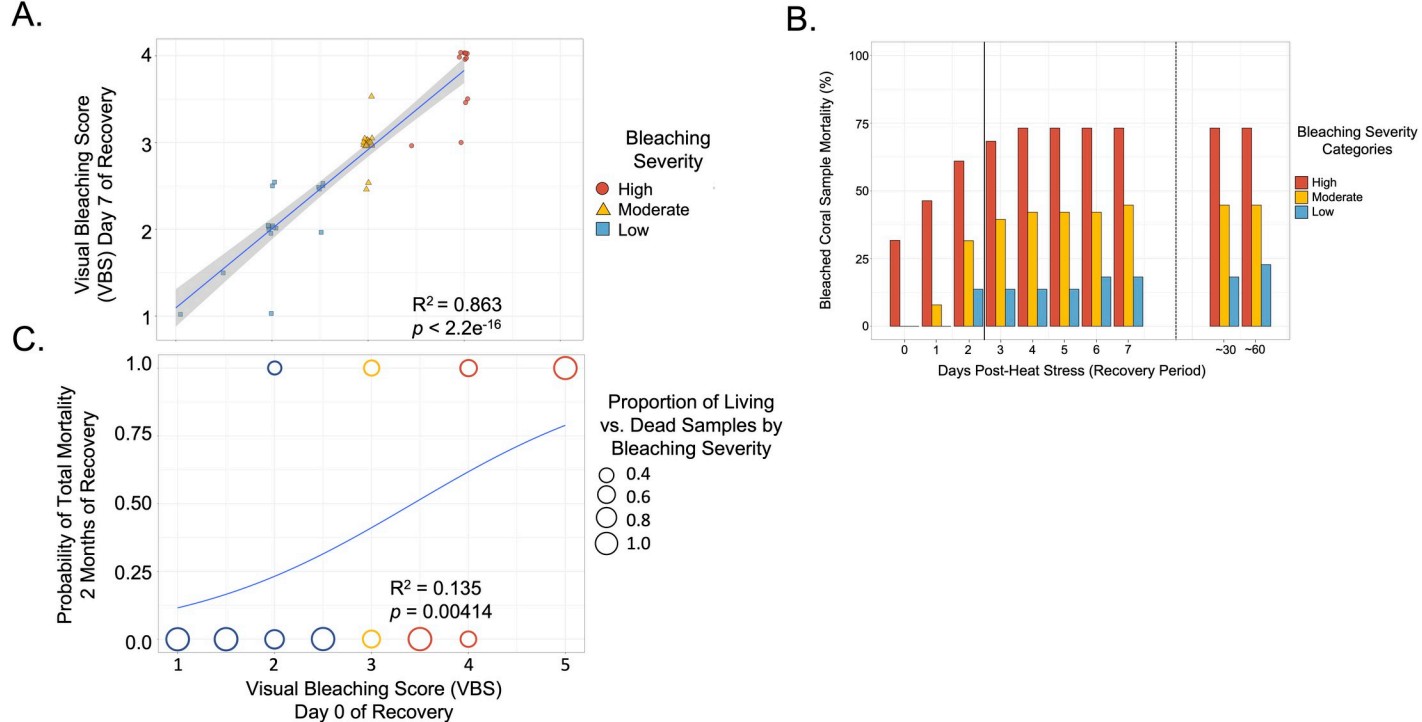

**Fig 3. Predictability of bleaching severity immediately after heat stress (Day 0 post-stress) for survival.** (A) Scatterplot showcasing sample bleaching severity on Day 0 versus Day 7 post-stress (based on visual bleaching score, VBS, and with bleaching severity categories), including a line of best fit with 95% confidence intervals (linear regression). A linear mixed effects model was used, with reef region as a random intercept to account for spatial variability, and the marginal $R^2$ value is provided. (B) Percentages, per bleaching severity group, of heat stressed genets with total tissue mortality daily from Day 0 to Day 7 post-stress and at the one month (~30 days) and two months (~60 days) timepoints. Mortality during the heat stress experiment (n = 13 genets) was attributed to the high bleaching severity category (i.e., low heat resistance), and severely bleached genets that died throughout the post-stress week were added to this count. The solid vertical line indicates the day by which 50% of all observed mortality was surpassed, and the dotted vertical line separates the short term (Days 0–7) and longer term (Days ~30 and ~60) post-stress periods. (C) Binomial representation of living (value = 0) and dead (value = 1) genets 2 months after heat stress versus bleaching severity on Day 0 post-stress, where circle sizes represent proportion of living vs. dead genets at each VBS category (mixed effects logistic regression, accounting for spatial variability, and the pseudo $R^2$ value is provided). Circle colors correspond to bleaching severity immediately after heat stress: blue = low bleaching, yellow = moderate bleaching, and red = high bleaching.

bleaching originated from ten of 21 patch reefs, a slightly lower proportion than the fore reef sites, where four (of seven) exhibited only moderate or worse bleaching (Fig 4A).

Cornwell et al., (2021) related bleaching differences on Palauan reefs to their respective temperature regimes, so we used our independent data to assess whether temperature extremes (i.e., recorded 10-minute interval events above 31˚C) of the originating reefs predicted how corals were able to survive heat stress [13]. We found a weak negative relationship between the three bleaching severity categories (Low, Moderate, and High) immediately after heat stress and reef temperature, in which only the low and moderate bleaching severity categories significantly differed (ordinal logistic regression, $p < 0.0001$). Genets that died during heat stress also tended to originate from cooler reefs, though further study is needed to address whether reef temperature strongly influences coral heat stress responses in this system (Fig 4B). Additionally, there was a large amount of variation in the reef specific data. For example, the four reef sites that had only severely bleached coral genets ranged in temperature extremes from well below (495.5, PR 33) to above the reef mean counts above 31˚C (2652.3, PR 14). Similarly, there was one reef that had entirely minimally bleached coral genets (Northern fore reef site 61) but a moderate count of temperature extremes (1526) (S1 Table). This suggested a

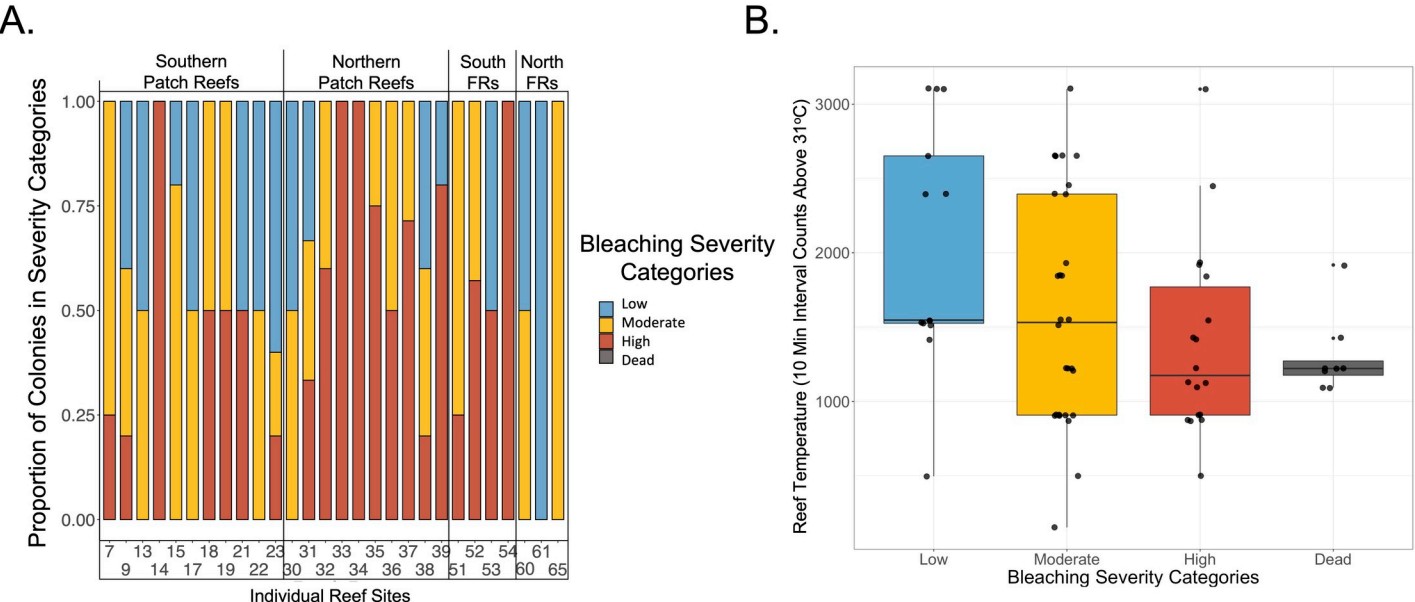

**Fig 4. Bleaching severity across variable reef environments and thermal regimes.** (A) Stacked barplot showing proportion of bleaching severity categories (low, moderate, and high bleaching) across the twenty-eight sampled reef sites. Reefs are organized by the following general groups: southern and northern patch reefs, and southern and northern fore reefs (written as FRs). (B) Boxplot showing the relationship between reef temperature extremes (represented by 10-minute interval HOBO logger counts above 31°C) and bleaching severity groups (low, moderate, and high bleaching). An ordinal logistic regression was performed on increasing bleaching severity groups (here from low severity to dead), whereby only the low and moderate bleaching severity groups differed significantly ($p < 0.05$).

general link between reef temperature and heat stress resistance but also highlighted that there are likely other environmental factors influencing resistance.

## Health beyond one week post bleaching

Because visual bleaching recovery did not occur in samples over the first week, we returned at one- and two-months post-stress to evaluate bleaching and mortality in the remaining samples. Out of the fifty surviving coral genets, thirty-seven achieved full visual recovery (VBS 1, 74%) after approximately one month. The remaining genets also all fell under the low bleaching severity category with minimal observed bleaching. There were representatives from all three original bleaching severity categories (Fig 5). After approximately two months, forty-nine out of fifty surviving heat stressed genets were still alive, and all had recovered fully (VBS 1) (Fig 5). All genets had a high probability of surviving and visibly recovering by two months post-bleaching if they reached Day 7 of the post-heat stress period. These collective results showed that bleaching sustained during short-term heat stress did play a significant role in the likelihood of survival past one week, though there are likely other important factors apart from bleaching severity to consider (Figs 3C and 5 and S4).

## Discussion

We tested how *Acropora hyacinthus* corals respond to short-term heat stress by assaying bleaching severity and mortality over one week post-stress and after one and two months post-stress. This study supports the increasingly popular use of short-term heat stress experiments [15, 16, 18, 41, 42] by providing detailed tracking of the stress and recovery response of corals to such experiments. Here, pigmentation loss in corals was highly stable under lab observation from 16 hours (i.e., the following morning after stress) to seven days post-stress. In addition,

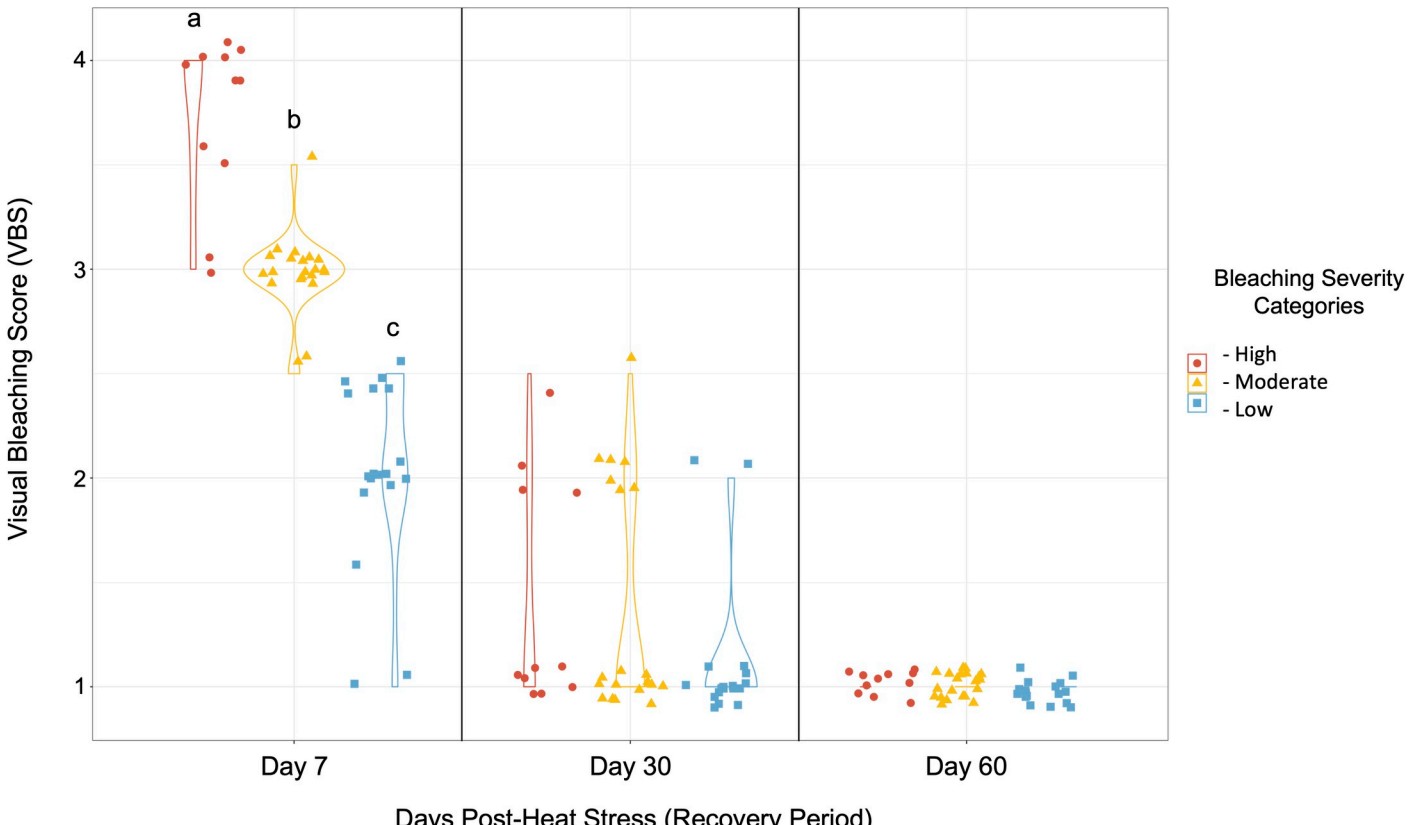

**Fig 5. Visual bleaching recovery beyond one week after heat stress.** Each point per panel denotes a coral genet (n = 50), and genets that died by Day 7 of the recovery period (n = 51) were excluded. Bleaching severity was measured using visual bleaching scores (VBS) at approximately Month 1 and Month 2; see S1 Table for the complete list of exact days for each fragment. Note that there were 12 genets (designated by a half VBS score) for which the two observers could not decide on a single bleaching category (e.g., VBS 1.5—between VBS 1 and 2). Tukey tests were performed on groups at each timepoint, with a significant threshold of $p \leq 0.05$. Bleaching severity between all resistance groups on Day 7 post-stress differed significantly where $p < 0.00001$, but not on Months 1 and 2.

corals that survived seven days after heating had high survival and recovery after 1–2 months in the lab. These data also confirm that bleaching is consistently stable days after a short-term heat stress experiment concludes and begin the process of ground-truthing whether short-term bleaching experiments can predict longer term survival. This experimental design also allows for testing links between variation in heat resistance and recovery. Further, we confirmed widespread variation in heat resistance and recovery of corals from variable thermal regimes and environments throughout Palau [see also 13].

### Reliability and efficiency of a non-destructive visual bleaching score metric

We utilized a five-point visual bleaching scoring metric—(1) none (2) visible, (3) moderate, (4) severe, and (5) total bleaching—in order to quickly and effectively evaluate a large sample size of control and heat stressed coral fragments. Visual bleaching scores were advantageously nondestructive and nondisruptive, meaning we did not sacrifice any portion of tracked colony fragments or disturb fragments to assess bleaching during the post-stress period. Several studies have previously relied solely on qualitative visual scoring methods [43–46] while others have calibrated visual results with quantitative assays like symbiont concentration via flow cytometry or sequencing and chlorophyll concentration [5, 13, 31, 35, 47]. Similarly, we confirmed our visual bleaching scores with flow cytometry data using ramet replicates sacrificed

after the heat stress experiment. We found that flow cytometry results matched well with the three main bleaching severity categories established with visual scoring (VBS 1–2, none to visible bleaching, i.e., highly resistant; VBS 3, moderate bleaching, i.e., moderately resistant; and VBS 4–5, severe to total bleaching, i.e., low resistant). However, we were unable to distinguish between some individual categories along the 5-point scale: none vs. visible and severe vs. total categories had similar symbiont concentrations, which suggests that visual scoring with the naked eye may not reliably distinguish between small pigmentation differences. Previous studies have shown that pigmentation loss may also correlate with chlorophyll and/or symbiont concentration depending on the individual [48] or species [49–51] and healthy coral colonies can have highly variable photosynthetic pigment concentrations [52]. Therefore, starting pigmentation or chlorophyll vs. symbiont concentration differences may have influenced quantitative correlation with qualitative visible bleaching in these *Acropora hyacinthus* colonies. Overall, confirming our qualitative results with a quantitative method allowed us to go forward with broad bleaching severity categories and showcased the importance of combining methodologies when evaluating bleaching. Together, this validated the merit of employing a reliable visual scoring metric in place of more expensive and time-consuming methods such as flow cytometry when performing experiments that require large sample sizes.

### Post bleaching trajectories are highly stable among corals with variable bleaching resistance

We found high intraspecific variation in bleaching sensitivity after the short-term heat stress experiment, comparable to other studies in which colonies were exposed to high temperatures during short-term [e.g., 13, 34] and long-term [e.g., 53] heat stress experiments. One week after heat stress, visual bleaching scores for coral fragments were not significantly different than they were after 16 hours (Fig 2). Among all corals that survived the first week after heat stress, 90% (n = 45) remained in the same bleaching severity category and 4 out of 5 of the other corals improved bleaching categories by Day 7 vs. Day 0. The lack of further visible symbiosis breakdown after heat stress suggested that symbiont loss was mostly restricted to the discrete heat stress event, while any remaining symbionts maintained an association with their host during the seven-day post-heat stress period. However, these remaining symbionts' systems such as photosynthetic ability and translocation of energy to the host may be significantly impaired during early holobiont stress recovery [54–56].

Mortality also followed these patterns. We found that 90% of mortality occurred within the first few days after short-term heat-stress. Mortality was high among corals with high bleaching within the first 3 days of heat stress (56%) but was low thereafter (another 7%). Mortality was far lower for corals with low or moderate bleaching (30%) and dropped to near zero after Day 3 (Fig 2). Higher mortality in corals with severe bleaching is consistent with what has been shown in a recent long-term study [57].

### Predictability of survival beyond one week after heat stress

We recorded a total mortality rate two months after short-term heat stress (~51%) that was comparable to studies *on Montipora capitata* (e.g., 60%) [48] and other bleaching susceptible species [58, 59]. However, our serial monitoring of time points showed that mortality visible after two months actually occurred within 3 days of bleaching for our system. One month after heat stress, we recorded that all corals were in the low bleaching severity category and the majority were fully recovered. Corals of all resistance types were among those that still had some visible bleaching after one month. We found high survival and full recovery among all surviving fragments two months after heat stress, even for those that had been severely bleached.

These data also show a significant correlation between survival after one week and likelihood of visible recovery in the future, which suggests that the first week after bleaching may be a critical period for short-term heat stressed corals. Indeed, 54% of the severely bleached corals died within three days of the heat stress experiments, compared to an overall 61% mortality of severely bleached corals two months after heat stress. By contrast, corals that were only minimally bleached in our stress tests experienced little mortality within three days (14%, n = 3 out of 22) and two months post-heat stress (23%, n = 5 out of 22). Similar results have been shown in field surveys of corals days to months after natural bleaching events [60–62] and report that severely bleached, white colonies seldom survive.

Here, potential limitations in the experimental design may have also impacted longer term bleaching and mortality results—for example, variation in symbiont concentration that may have been masked by visually scoring bleaching severity [13, 47], slightly lower light intensity levels of the heat stress tank system compared to natural sunlight conditions in the different originating reef environments, and the lack of supplemental feeding during longer term heat stress and bleaching recovery [17].

## Intraspecific variation in thermal resistance and recovery throughout Palau

We found that high bleaching resistance was geographically widespread across variable environments though more common in corals coming from hotter reefs, with similar results reported in Cornwell et al., (2021) [13]. A minority of reefs had corals of all bleaching resistant types. We report that half of all reefs had highly resistant corals, while 71% of all reefs each had low resistant and/or moderate resistant corals. Furthermore, reef temperature extremes were weakly associated with bleaching severity [e.g., 10, 19], and with post-stress mortality [e.g., 63].

We also report for the first time that recovery from severe bleaching may also be a widespread trait for *Acropora hyacinthus* in Palau: coral genets from all geographic regions survived and had visible recovery in our experimental system. We recorded 100% mortality of bleached coral genets from only six out of the 28 reef sites. This suggests that corals with high bleaching recovery ability may be located throughout Palau. It is also possible that other facets of environmental variation could similarly impact heat resistance and recovery. For example, intensity and duration of variability, accumulated heat stress (e.g., degree heating weeks), availability of nutrients, and prevalence of diseases and predators may play a more consequential role predicting the likelihood of heat resilience and survival [64–67].

## Implications and future directions

Short-term heat stress experimental designs (reviewed in McLachlan et al., 2020) have been employed as a powerful tool to globally and rapidly assess susceptibility to bleaching and likelihood of survival in susceptible corals [27]. An advantage of this approach is the ability to test many coral genets rapidly within 2–3 days of collection [20, 22, 68, 69]. This minimizes experimental effects such as acclimation to lab conditions and reduces the chance of other sources of mortality or stress such as starvation. However, these advantages must be weighed against the fact that minimal post-collection recovery time could confound transportation and heat stress, and the strong heat pulse could impair coral function so quickly that it might take days or longer for the full reaction to become apparent [17]. In our dataset, non-stressed control fragments were highly robust, delayed reactions among heat stressed fragments were rare, and rapid assays of bleaching were accurate measures of longer-term response. Other potential disadvantages are that standardized experimental heat stress applied to corals from variable reef environments may not fully capture the natural variability these corals experience nor the

spectrum of natural bleaching events that are known to vary in duration and intensity. This study can be built upon by incorporating experimental components that further mimic natural reef environments (supplemental feeding, light conditions, baseline and ramp temperature regime, and available nutrients) and by outplanting heat stressed coral fragments to a reef environment to better relate short-term heat stress recovery results in a controlled system to ecological recovery. It is imperative to combine reliable standardized protocols, which can reveal initial integral mechanistic patterns, with studies that integrate other environmental factors and variable durations and intensities of heat stress.

This study was conducted solely on the bleaching sensitive species *Acropora hyacinthus*. Species in this genus react quickly to strong heat stress, whereas other genera such as *Pocillopora* [70, 71] or *Porites* [72] may require longer exposure to lower levels of heat to bleach without immediate death. Another important future direction is to use this experimental design on other prominent reef-building coral species and in other variable reef types. Future assays should also test physiological (e.g., metabolic baseline, heterotrophy, and symbiont resilience), and genomic predictors of thermal resilience during this short time period—as these factors are known to relate to thermal sensitivity over longer timescales [32, 51, 73–78].

## Conclusions

We have used a simple, rapid, and low-cost experimental design to test corals for bleaching resistance and recovery. This coupling is important, because resilience against ocean warming may require a combination of resistance and recovery [79, 80]. Our data confirm that bleaching and mortality recorded quickly after short-term heat stress experiments are stable and reliable measures of coral stress phenotypes. In addition, our findings that high heat resistance and recovery may be widespread throughout Palau and that low resistant coral genets can also recover well suggest that many *Acropora hyacinthus* individuals may have the metabolic machinery necessary to effectively resist and/or recover from heat stress. Ultimately, prioritizing the highest bleaching resistant corals may help maintain resilience on reefs in the future [81], but including corals of variable resistance and recovery ability in reef management plans may further increase diversity and sustainability.

## Supporting information

**S1 Fig. Geographic locations of all reefs and reef groups.** Map of Palau with relative reef positions, created using R packages rgeos, mapdata, and rgdal and the "Palau_Shoreline" shapefile from USGS. Reefs are represented by orange circles and grouped based on reef clusters; patch reefs are circled in red, yellow, green, and blue according to their region, and fore reef sites are circled in purple. Latitudinal and longitudinal coordinates of all reefs are included and written in decimal degrees.
(TIF)

**S2 Fig. Accuracy and precision of the heat stress tank temperature ramp system.** Representation of the temperature ramp system in a heat stress tank outfitted with a heater, chillers, a light fixture, and water inflow and outflow tubing as described further in the methods section. Real-time temperature measurements were collected using a HOBO logger recording temperature in 10-minute intervals. This example temperature ramp was measured on July 5th 17:20 until July 6th 09:00 2022. The ramp up period (31–34˚C) was programmed for 2.5 hours, the hold period (34˚C) was programmed for 2.5 hours, and then the ramp down and hold at 30˚C was programmed for 10 hours and 40 minutes. Note that this example temperature ramp system differed from the ramp system conducted for analysis (3 hour ramp up from 30 to 34.5˚C,

3 hour hold at 34.5˚C, and ramp down to 30˚C) but these measurements aimed to demonstrate the high degree of accuracy and precision of the temperature controller system.
(TIF)

**S3 Fig. Bleaching resistance variation following heat stress.** Counts of control (blue) and bleached (green) colony samples before and after the two-day short-term heat stress experiment. Bleaching severity before and after heat stress was measured by visual bleaching score (VBS).
(TIF)

**S4 Fig. Relationship between bleaching severity immediately after heat stress (Day 0 post-stress) and mortality on Day 7 and Month 1 post-stress.** Binomial representation of living (value = 0) and dead (value = 1) coral fragments 7 days (A) and 1 month (B) after heat stress versus bleaching severity on Day 0 post-stress, where circle sizes represent proportion of living vs. dead samples at each VBS category (mixed effects logistic regression, accounting for spatial variability, with pseudo $R^2$ values provided). Note that identical statistics results from both logistic regressions are due to no corals dying between Day 7 and Month 1 timepoints. Circle colors correspond to bleaching severity immediately after heat stress: blue = low bleaching, yellow = moderate bleaching, and red = high bleaching.
(TIF)

**S5 Fig. Breakdown of bleaching severity across geographic locations and reef temperatures.** (A) Violin plot of bleaching severity and geographic locations. Bleaching severity is measured through visual bleaching scores (VBS 1, no bleaching, to VBS 5, total bleaching), and each point denotes a fragment that represents a coral colony within geographic locations. All locations are in lagoons apart from the Fore Reef category, and colors correspond to groups in S1 Fig. We ran an ANOVA and Tukey test for significance (S2 Table), * denotes $p \leq 0.05$. (B) Scatterplot showing the relationship between reef temperature extremes (represented by 10-minute interval HOBO logger counts above 31˚C) and visual bleaching score with 95% confidence intervals. Day 0 and 7 after heat stress are shown. Linear regression results: Day 0, $p = 0.02167$, $R^2 = 0.0731$, df = 57, and Day 7, $p = 0.0292$, $R^2 = 0.1287$, df = 28.
(TIF)

**S6 Fig. Photographic coral fragment representations of the 5 visual bleaching score categories and total mortality.** Two observers determined each fragment's visual bleaching score category and mortality. Here, one fragment example is provided per visual bleaching score category and mortality category.
(TIF)

**S1 Table. Metadata spreadsheet.** Visual bleaching scores, symbiont proportion, temperature data, descriptions of reef and resistance/bleaching severity groupings, and colony identifications.
(XLSX)

**S2 Table. Summary of statistics results.** All statistics result outputs, including linear regressions, logistic regressions, ANOVAs, and post-hoc Tukey tests.
(XLSX)

## Acknowledgments

We thank the staff and boat operators at the Palau International Coral Reef Center, and Stanford University undergraduate interns Callan Hoskins, Colin Hyatt, Mehr Kumar, and Julien

Ueda for their assistance collecting samples and running experiments in the field. We also acknowledge current and former members of the Palumbi lab—Elora López-Nandam, Erik Hanson, Kristin Robinson, and Marilla Lippert—as well as Andrea G. Grottoli, Elizabeth Hadly, John R. Pringle, and referees during the review process for their comments and edits on the manuscript.

## Author Contributions

**Conceptualization:** Nia S. Walker, Stephen R. Palumbi.

**Data curation:** Nia S. Walker, Brendan H. Cornwell, Victor Nestor, Katrina C. Armstrong.

**Formal analysis:** Nia S. Walker.

**Funding acquisition:** Yimnang Golbuu, Stephen R. Palumbi.

**Investigation:** Nia S. Walker, Brendan H. Cornwell, Victor Nestor, Katrina C. Armstrong.

**Methodology:** Nia S. Walker, Brendan H. Cornwell, Katrina C. Armstrong, Stephen R. Palumbi.

**Resources:** Yimnang Golbuu.

**Supervision:** Stephen R. Palumbi.

**Visualization:** Nia S. Walker.

**Writing – original draft:** Nia S. Walker.

**Writing – review & editing:** Nia S. Walker, Brendan H. Cornwell, Victor Nestor, Katrina C. Armstrong, Yimnang Golbuu, Stephen R. Palumbi.

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
