## [Decision Letter · Decision Letter 0]

28 Jun 2022

PONE-D-22-13834Persistence of phenotypic responses to short-term heat stress in the tabletop coral Acropora hyacinthusPLOS ONE

Dear Dr. Walker,

Thank you for submitting your manuscript to PLOS ONE. After careful consideration, we feel that it has merit but does not fully meet PLOS ONE’s publication criteria as it currently stands. Therefore, we invite you to submit a revised version of the manuscript that addresses the points raised during the review process.

We look forward to receiving your revised manuscript.

Kind regards,

Christian R. Voolstra, Ph.D.

Academic Editor

PLOS ONE

Journal Requirements:

3. We note that Figure S1 in your submission contain map/satellite image which may be copyrighted. All PLOS content is published under the Creative Commons Attribution License (CC BY 4.0), which means that the manuscript, images, and Supporting Information files will be freely available online, and any third party is permitted to access, download, copy, distribute, and use these materials in any way, even commercially, with proper attribution. For these reasons, we cannot publish previously copyrighted maps or satellite images created using proprietary data, such as Google software (Google Maps, Street View, and Earth). For more information, see our copyright guidelines: http://journals.plos.org/plosone/s/licenses-and-copyright.

a. You may seek permission from the original copyright holder of Figure S1 to publish the content specifically under the CC BY 4.0 license.  

Reviewers' comments:

Reviewer's Responses to Questions

**Comments to the Author**

1. Is the manuscript technically sound, and do the data support the conclusions?

Reviewer #1: Yes

Reviewer #2: Yes

2. Has the statistical analysis been performed appropriately and rigorously? 

Reviewer #1: Yes

Reviewer #2: Yes

3. Have the authors made all data underlying the findings in their manuscript fully available?

Reviewer #1: Yes

Reviewer #2: Yes

4. Is the manuscript presented in an intelligible fashion and written in standard English?

Reviewer #1: Yes

Reviewer #2: Yes

5. Review Comments to the Author

Reviewer #1: Review of manuscript PONE-D-22-13834 – “Persistence of phenotypic responses to short-term heat stress in the tabletop coral Acropora hyacinthus”

Main Comments

The authors conducted a series of short-term heat stress experiments to assess the thermal tolerance and associated recovery dynamics for a number of Acropora hyacinthus colonies in Palau, Micronesia. Importantly, the authors tracked the recovery dynamics of corals following heat stress in detail, complementing recent studies assessing differences in stress resistance of corals to acute thermal stress. This study is of value to the field and the paper is clear and well-written, though there are some aspects of the analysis and interpretation and presentation of the results that should be addressed before the manuscript is ready for publication.

More specific, line-by-line comments and suggestions are included below.

Specific comments:

Abstract:

Line 35: This is a little confusing to read – would suggest changing to ‘… by monitoring the phenotypic response and recovery of fragments from 101 Acropora hyacinthus colonies in Palau, Micronesia, to short-term heat stress.’

Line 38: This also reads a little awkwardly – suggest changing to ‘…and after one and two months of recovery.’

Introduction:

Line 72: I think here and throughout, you should be citing actual short-term heat stress experimental studies rather than a meta-analysis that mentions short-term heat stress studies (i.e., the Grottoli paper).

Line 77-78: I think there’s comma missing somewhere here, maybe after ‘stressors’?

Line 87: ‘mindset’ seems a little anthropomorphic. Would suggest changing to ‘…from a stressed to a recovery state.’

Line 94: Suggest changing to ‘… phenotypic responses of coral fragments to heat stress…’

Line 101-102: ‘the experimental system’ seems like an unusual way to phrase it. Why not just say ‘the coral fragments’?

Methods:

Line 126: Can you provide a little more detail here, or in a table format, regarding the experiments? Information such as the number of nubbins in a tank at a given time and if some of the nubbins going through day 1 of the heat ramps overlap with some from day 2? Was any of this accounted for statistically regarding potential batch effects?

Line 132: Would suggest changing ‘on reefs’ to ‘on each reef’

Line 199-200: What inflow does ½ volume/h actually equate to?

Line 203-206: Could you provide figures/data regarding the accuracy and precision of the temperature treatments?

Line 208-210: I think this statement should come after lines 212-213 where you mentioned sacrificing one replicate for symbiont counts.

Line 212: For complete clarity, would be good to add ‘one’ in front of ‘control nubbin’

Line 218: ‘were used’ for what? For long-term monitoring of recovery? And how many holding tanks were used?

Line 219-222: Is there any more detailed data on temperature and light levels in the holding tanks? How similar/different were conditions between holding tanks?

Line 257-258: The statement regarding reef regions seems like a result rather than methods?

Results:

Line 267: here and throughout, it would be good to report p-values to support statements like ‘were highly distinct’

Line 342: I think ‘88’ can be written numerically

Line 353-355: This statement is confusing

Line 388: I’m not sure this statement should start with ‘However’ or include ‘also’, I think something like ‘Overall’ would be more appropriate as this statement doesn’t go against the previous statement, but rather describes a different aspect of the data.

Line 405: Should this refer to Fig. 5 and not Fig. 4?

Discussion:

Line 423-424: See my comment earlier about the Grottoli paper. Also, I’d suggest rephrasing this statement as it isn’t very clear. ‘commonly used’ is vague and ‘that are normally assayed within 16h post stress is oddly specific. Can you generalise this to highlight that this study complements the increasingly popular use of short-term heat stress experiments by providing detailed tracking of the stress and recovery response of corals to such experiments

Line 431: maybe add ‘of corals from’ after ‘recoverability’?

Line 448: ‘none to visible bleaching’ isn’t very clear – could this be rephrased for clarity? Also, should ‘high resistant’ be ‘highly resistant’?

Line 468: ‘similar and variable lengths of time’ is vague – expand on this a little?

Line 496: suggest adding ‘surviving’ between ‘all nubbins’

Line 501-502: could you add percentages to support ‘most of the corals’ and ‘virtually all the corals’?

Line 506: Could the low amount of variation explained also be due to the course bleaching categories used? Some classed as having a low VBS may in fact have been worse off than visibly judged? In general, I think the Discussion talks about needing to consider other environmental or biotic drivers multiple times (e.g., here and lines 527-530 below), but doesn’t consider potential limitations of the experimental and statistical approach and how this could have affected the accuracy of the study

Line 534: The McLachlan paper isn’t a short-term heat stress experiment, suggest citing actual examples of studies using these experiments – especially if there are studies out there testing corals ‘in a variety of reef types’ as you state

Line 544: ‘such delayed’ is confusing, should it just be ‘delayed’

Line 561: While I agree with this statement, part of it feels a little repetitive of prior statements lines 506 and 527. Also, if you do keep this, maybe reduce the number of examples in each set of parentheses as this sentence is very long

Line 575: Could change ‘of the stress reaction’ to ‘of coral stress phenotypes’? Just a suggestion

Figures:

Figure 2: Is there a clearer way to present this? It isn’t the easiest figure to follow

Figure 5: Could you edit this figure so that the individual points are aligned with each violin? Also in the figure legend, line 415 – ‘with in between VBS scores’ reads awkwardly, would suggest rephrasing

Supplement:

Figure S3: I think there’s an error with R2 and p-values as they’re the same in both panels

Reviewer #2: General Comments:

The authors present a comprehensive analysis of how immediate responses of corals to short-term (48hr) heat-stress correspond to responses over a week and one and two months post stress. The study is well designed, the data are clearly presented, and the conclusions are generally well-suited to the data. One nuance that may be worth spending a little more time discussing is the fact that the experimental design does not allow for examination of how "recovery" as examined in this experiment relates to in situ/ecological recovery from a natural bleaching event. Here, the data show clearly and convincingly that some corals are capable of recovering from a short-term stress exposure, which in and of itself is an interesting finding. However, without comparison data from a longer-term experiment or natural bleaching event it is difficult to draw conclusions on how the patterns of recovery herein might relate to natural recovery. To be fair, I don't think the authors have gone too far in interpreting their results, only that another sentence or two could be devoted to discussing how to build on these current results to better relate short-term recovery to ecological recovery (perhaps in the future directions section).

Aside from a few minor technical and grammatical comments noted below which should be easily dealt with in a minor revision, I support acceptance and publication of this manuscript.

Sincerely,

dan barshis

Line by line comments:

Introduction:

Line 64, species "composition" or "identity"?

Lines 86-87, maybe change to "... stressed to a recovery state" as coral mindset seems a bit anthropomorphic

Line 94, "In this study[,]"

Methods:

I suggest considering use of the term "ramet" instead of "nubbin" as ramet additionally specifies that nubbins are from the same parent colony vs. nubbins could be from a mix of colonies. I've moved to using "colony" and "ramet" unless I know for sure that "colonies" are unique "genets" but it's up to you if you want to stick with the current terms.

I also highly recommend including a github repository with all of your R code so that people can properly recreate your analyses.

Line 168-169/182-183, do you have a citation for the identified symbiont cells divided by total cell counts methodology? I'm a little unclear on the details of the calculation. Wouldn't a better ratio be the number of symbiont cells to non-symbiont (i.e., host) cells? I can see how the symbiont/total cell ratio would be somewhat proportional to symbiont cell density per unit host tissue, however, with this ratio the relationship would not be linear correct? Consider the hypothetical scenario where you have two identically sized coral fragments (i.e., same amount of host tissue) and one has many symbionts and one is mostly bleached. In this case the bleached one would have a very low symbiont/total cell ratio while the densely populated one would not have as high a ratio as it should because the total cell number (denominator) is greatly increased by the number of symbiont cells. I might be wrong here but either a citation comparing this metric with more commonly used metrics or some additional acknowledgement of the limitations of this method would be useful. I think as a rough proxy for actual cell densities it should be fine but would be important to acknowledge the caveats so that other groups don't naively take it up as a direct replacement for a true symbiont density per host biomass measure.

Line 202, please include what kind of PAR sensor was used to measure the light levels (e.g., planar or spherical) as this affects the values recorded. Also, these light levels are quite low compared to average PAR values on the reef. How might this have affected your results?

Lines 220-222, were light levels in the holding tanks ever recorded? Would be curious how they compared to your experimental tank light levels.

Lines 242-244, how was an ANOVA used to determine correlation, looks like it was the lme model maybe that generated the correlation in Table S2 not the ANOVA?

Line 244, please specify factors included in the lme model (i.e., what was included as fixed versus random effects)

Line 261, suggest adding a sentence to the tune of "Specific model formulas and outputs can be found in Table S2."

Results:

Lines 264-279, suggest including the R2 here as well as other relevant results of the statistics in Table S2. Wording such as "highly distinct" or "little distinction" would be better clarified with p-values.

Lines 283-290, suggest specifying how this wording corresponds with your numerical categories here to avoid the reader having to flip back and forth to see how they align.

Line 314, please clarify whether this figure includes only heat treated or heat treated and control fragments.

Line 328, I suggest including the 1 mo and 2 mo timepoints on here with a broken x-axis. The mortality data aren't really presented elsewhere in graphical form. You could also look into a cox proportional hazards/survival curve approach/figure instead as that is more common for this kind of mortality over time data.

Lines 353-355, I'm a bit confused on what data this last sentence is referring to. Above you discuss Day 0 data vs. Day 2/Day 7/Day 60 but I don't see non-Day 0 vs. Day 7 comparisons?

Lines 360, "coincide[d]" no?

Lines 404, suggest changing to "... forty-nine out of fifty [surviving] heat stressed nubbins ..."

Lines 424-425, suggest adding "[reviewed in]" in front of the Grottoli reference.

Lines 426-427, why 3 days here when above you focus on day 7?

Line 431, what's the difference between recoverability and recovery? If none, suggest sticking with recovery to avoid introducing another term.

Lines 455-457, could be varying starting pigmentation or it could just be the naked eye is unable to reliable distinguish between categories 1 vs. 2 and 4 vs. 5. Suggest adding wording to this effect.

Lines 471, I was a little confused when I first read this sentence. Suggest adding "improved bleaching categories [on day 7 vs. day 0]."

Lines 499-506, Either here or earlier on, I suggest referencing Evensen et al_2021_Limnology and Oceanography_Remarkably high and consistent tolerance of a Red Sea coral to acute and chronic thermal stress exposures as showing that both bleaching and sub-bleaching physiological responses can be consistent between very short and medium term stress exposures.

Lines 522-530, I'm missing some discussion of how your in-tank survival/recovery might relate to in situ recovery/survival. I think it would be best in this paragraph simply to acknowledge, remind the reader that your experiment looked at recovery in a controlled environment, and that recovery may have looked different if your fragments had been returned to the reef.

6. PLOS authors have the option to publish the peer review history of their article (what does this mean?). If published, this will include your full peer review and any attached files.

Reviewer #1: No

Reviewer #2: **Yes: **Daniel Barshis

---

## [Author Response · Author response to Decision Letter 0]

11 Aug 2022

Dear Editor and Reviewers, in this “Response to Reviewers” document you will find explanations of all revisions. We have provided a brief summary below of main revisions requested, and then we also address every comment. Thank you for the suggestions, which we believe have improved the manuscript for publication. Responses to each reviewer comment begin with a “>” symbol.

Main revisions requested:

1. We confirmed that the manuscript and all additional files meet PLoS ONE’s style requirements.

2. We added information regarding all the collection and export permits for our work, including national and state Palau permits and a CITES permit. Information is described in the first Materials and Methods paragraph.

3. It was brought to our attention that the old Figure S1 may have a copyright issue. We remade the figure in R to avoid any copyright issue. The new Fig S1 was made using R packages rgeos, mapdata, rgdal, and the “Palau_Shoreline” shapefile from USGS, described further in the S1 Fig caption.

4. Per reviewer recommendation, we clarified details in the heat stress tank system setup in the methods and acknowledged caveats and limitations in the discussion. We also edited Figures 2, 3, and 5. 

5. Reviewers requested additional citations of short-term heat stress studies throughout the manuscript, and we added more references for this. We also removed sentences and references in the Discussion section that reviewers suggested were redundant.

a. Added references:

1. Evensen NR, Fine M, Perna G, Voolstra CR, Barshis DJ. Remarkably high and consistent tolerance of a Red Sea coral to acute and chronic thermal stress exposures. Limnology and Oceanography. 2021;66(5): 1718-1729. doi: 10.1002/lno.11715. 

2. Savary R, Barshis DJ, Voolstra CR, Meibom A. Fast and pervasive transcriptomic resilience and acclimation of extremely heat-tolerant coral holobionts from the northern Red Sea. PNAS. 2021;118(19): e2023298118. doi: 10.1073/pnas.2023298118.

3. Seneca FO, Palumbi SR. The role of transcriptome resilience in resistance of corals to bleaching. Molecular Ecology. 2015;24(7): 1467-1484. doi: 10.1111/mec.13125

4. Cunning R, Baker AC. Thermotolerant coral symbionts modulate heat stress-responsive genes in their hosts. Molecular Ecology. 2020;29(15): 2940-2950. Doi: 10.1111/mec.15526

5. Voolstra CR, Valenzuela JJ, Turkarslan S, Cardenas A, Hume BCC, Perna G, et al. Contrasting heat stress response patterns of coral holobionts across the Red Sea suggest distinct mechanisms of thermal tolerance. Molecular Ecology. 2021;30(18): 4466-4480. doi: 10.1111/mec.16064 

6. Leinbach SE, Speare KE, Rossin AM, Holstein DM, Strader ME. Energetic and reproductive costs of coral recovery in divergent bleaching responses. Scientific Reports 2022;11: 23546. doi: 10.1038/s41598-021-02807-w

b. Removed references:

1. Burkepile DE, et al. Nitrogen identity drives differential impacts of nutrients on coral bleaching and mortality. Ecosystems 2019;23:798-811. doi: 10.1007/s10021-019-00433-2

2. McClanahan TR, et al. Effects of geography, taxa, water flow, and temperature variation on coral bleaching intensity in Mauritius. MEPS 2005;298:131-142. doi: 10.3354/meps298131

3. Morris LA, et al. Nutrient availability and metabolism affect the stability of Coral-Symbiodiniaceae symbioses. Trends in Microbiology 2019;27(8):678-689. doi: 10.1016/j.tim.2019.03.004

4. Muthukrishnan R, Fong P. Multiple anthropogenic stressors exert complex, interactive effects on a coral reef community. Coral Reefs 2014;33:911-921. doi: 10.1007/s00338-014-1199-1

5. Roder C, et al. Metabolic plasticity of the corals Porites lutea and Diploastrea heliopora exposed to large amplitude internal waves. Coral Reefs 2011;30:57-69. doi: 10.1007/s00338-011-0722-x

6. Schoepf, V, et al. Thermally variable, macrotidal reef habitats promote rapid recovery from mass coral bleaching. Front Mar Sci 2020;7:245. doi: 10.3389/fmars.2020.00245

7. Wooldridge SA, Done TJ. Improved water quality can ameliorate effects of climate change on corals. Ecological Applications 2009;19(6):1492-1499. doi: 10.1890/08-0963.1

Journal Requirements:

>Thanks, we have confirmed we meet the style requirements.

>We have added the requested information about permits for our work, thanks.

3. We note that Figure S1 in your submission contain map/satellite image which may be copyrighted. All PLOS content is published under the Creative Commons Attribution License (CC BY 4.0), which means that the manuscript, images, and Supporting Information files will be freely available online, and any third party is permitted to access, download, copy, distribute, and use these materials in any way, even commercially, with proper attribution. For these reasons, we cannot publish previously copyrighted maps or satellite images created using proprietary data, such as Google software (Google Maps, Street View, and Earth). For more information, see our copyright guidelines: http://journals.plos.org/plosone/s/licenses-and-copyright.

a. You may seek permission from the original copyright holder of Figure S1 to publish the content specifically under the CC BY 4.0 license. 

>Thank you for pointing this out. To be conservative, we have recreated the map for Figure S1 using R packages (rgeos, mapdata, and rgdal) and using the shapefile from the Palau mapping from USGS. Our Fig S1 caption has been updated to reflect the change.

>Thanks, we have confirmed the reference list is up to date.

Reviewers' comments:

Reviewer's Responses to Questions

Comments to the Author

1. Is the manuscript technically sound, and do the data support the conclusions?

Reviewer #1: Yes

Reviewer #2: Yes

2. Has the statistical analysis been performed appropriately and rigorously?

Reviewer #1: Yes

Reviewer #2: Yes

3. Have the authors made all data underlying the findings in their manuscript fully available?

Reviewer #1: Yes

Reviewer #2: Yes

4. Is the manuscript presented in an intelligible fashion and written in standard English?

Reviewer #1: Yes

Reviewer #2: Yes

5. Review Comments to the Author

Reviewer #1: Review of manuscript PONE-D-22-13834 – “Persistence of phenotypic responses to short-term heat stress in the tabletop coral Acropora hyacinthus”

Main Comments

The authors conducted a series of short-term heat stress experiments to assess the thermal tolerance and associated recovery dynamics for a number of Acropora hyacinthus colonies in Palau, Micronesia. Importantly, the authors tracked the recovery dynamics of corals following heat stress in detail, complementing recent studies assessing differences in stress resistance of corals to acute thermal stress. This study is of value to the field and the paper is clear and well-written, though there are some aspects of the analysis and interpretation and presentation of the results that should be addressed before the manuscript is ready for publication.

More specific, line-by-line comments and suggestions are included below.

Specific comments:

Abstract:

Line 35: This is a little confusing to read – would suggest changing to ‘… by monitoring the phenotypic response and recovery of fragments from 101 Acropora hyacinthus colonies in Palau, Micronesia, to short-term heat stress.’

>Thanks, we made the change.

Line 38: This also reads a little awkwardly – suggest changing to ‘…and after one and two months of recovery.’

>Thanks we made the change.

Introduction:

Line 72: I think here and throughout, you should be citing actual short-term heat stress experimental studies rather than a meta-analysis that mentions short-term heat stress studies (i.e., the Grottoli paper).

>Thanks, we made the change to directly incorporate more short-term heat stress studies. 

Line 77-78: I think there’s comma missing somewhere here, maybe after ‘stressors’?

>Thanks, we made the change.

Line 87: ‘mindset’ seems a little anthropomorphic. Would suggest changing to ‘…from a stressed to a recovery state.’

>Thanks for the suggestion, we made the change.

Line 94: Suggest changing to ‘… phenotypic responses of coral fragments to heat stress…’

>Thanks we made the change.

Line 101-102: ‘the experimental system’ seems like an unusual way to phrase it. Why not just say ‘the coral fragments’?

>Thanks for catching this, we made the change.

Methods:

Line 126: Can you provide a little more detail here, or in a table format, regarding the experiments? Information such as the number of nubbins in a tank at a given time and if some of the nubbins going through day 1 of the heat ramps overlap with some from day 2? Was any of this accounted for statistically regarding potential batch effects?

>Thanks for the suggestions, we incorporated the changes. Nubbins at different stages of the heat stress experiment (i.e. day 1 vs. day 2) did not overlap in heat stress tanks. The number of nubbins ranged from 15-25 per heat stress tank. These nubbins were slotted into heat stress tanks with other nubbins not included in this manuscript’s analysis, as part of a larger heat resistance experiment survey (Cornwell et al., 2021). We did not statistically account for potential batch effects, due to the negligible mortality of our non-stressed controls across all tanks during the heat stress experiment.

Line 132: Would suggest changing ‘on reefs’ to ‘on each reef’

>Thanks we made the change.

Line 199-200: What inflow does ½ volume/h actually equate to?

>Thanks, we clarified that the heat stress tanks each held 10 liters.

Line 203-206: Could you provide figures/data regarding the accuracy and precision of the temperature treatments?

>Thanks for the question, we have now provided another supplemental figure showing a real-time measurement (via HOBO loggers) of an example temperature treatment within a day. Measurements fell within +/- 0.1 oC of the target holds of 34 oC and 30 oC throughout the entire period. This is now figure S2.

Line 208-210: I think this statement should come after lines 212-213 where you mentioned sacrificing one replicate for symbiont counts.

>We incorporated the change, thanks.

Line 212: For complete clarity, would be good to add ‘one’ in front of ‘control nubbin’

>Changed, thanks.

Line 218: ‘were used’ for what? For long-term monitoring of recovery? And how many holding tanks were used?

>Thanks for asking us to clarify, we addressed the confusing language. There was 1 holding tank used for all corals in recovery (control and heated). Partway through the recovery experiment, we transferred all corals to another comparable recovery tank to remove macroalgal buildup.

Line 219-222: Is there any more detailed data on temperature and light levels in the holding tanks? How similar/different were conditions between holding tanks?

>Thanks for the question. Unfortunately, we don’t have more detailed data on temperature and light levels. We continued to periodically monitor temperature in the recovery tanks throughout the experiment. The 2 holding tanks used throughout the experiment (1 before 9/10/18 and then the other 9/10-10/2/18) were adjacent and had comparable temperature recordings and access to light.

Line 257-258: The statement regarding reef regions seems like a result rather than methods?

>Thanks, we removed the statement.

Results:

Line 267: here and throughout, it would be good to report p-values to support statements like ‘were highly distinct’

>Thanks, we added the stats reference.

Line 342: I think ‘88’ can be written numerically

>Thanks, made the change.

Line 353-355: This statement is confusing

>Thanks, we agree and removed the statement.

Line 388: I’m not sure this statement should start with ‘However’ or include ‘also’, I think something like ‘Overall’ would be more appropriate as this statement doesn’t go against the previous statement, but rather describes a different aspect of the data.

>Thanks, we made the change.

Line 405: Should this refer to Fig. 5 and not Fig. 4?

>Yes, thanks very much for catching this.

Discussion:

Line 423-424: See my comment earlier about the Grottoli paper. Also, I’d suggest rephrasing this statement as it isn’t very clear. ‘commonly used’ is vague and ‘that are normally assayed within 16h post stress is oddly specific. Can you generalise this to highlight that this study complements the increasingly popular use of short-term heat stress experiments by providing detailed tracking of the stress and recovery response of corals to such experiments

>Thanks very much, we rephrased these opening sentences for more clarity and added the requested references.

Line 431: maybe add ‘of corals from’ after ‘recoverability’?

>Thanks, we made the change.

Line 448: ‘none to visible bleaching’ isn’t very clear – could this be rephrased for clarity? Also, should ‘high resistant’ be ‘highly resistant’?

>Thanks, we made the change! We were referring to the visual bleaching scores mentioned earlier in the paragraph, VBS 1 and 2 are highly resistant.

Line 468: ‘similar and variable lengths of time’ is vague – expand on this a little?

>Thanks, we made the change!

Line 496: suggest adding ‘surviving’ between ‘all nubbins’

>Thanks, we made the change.

Line 501-502: could you add percentages to support ‘most of the corals’ and ‘virtually all the corals’?

>Thanks, we made the change and rephrased the sentences.

Line 506: Could the low amount of variation explained also be due to the course bleaching categories used? Some classed as having a low VBS may in fact have been worse off than visibly judged? In general, I think the Discussion talks about needing to consider other environmental or biotic drivers multiple times (e.g., here and lines 527-530 below), but doesn’t consider potential limitations of the experimental and statistical approach and how this could have affected the accuracy of the study.

>Thanks for the suggestions. We have now included more caveats that address the experimental design. We meant here that a few corals with none to minimal bleaching (highly bleaching resistant) exhibited some mortality, and we suggested a few putative reasons for why these few highly bleaching resistant corals still died.

Line 534: The McLachlan paper isn’t a short-term heat stress experiment, suggest citing actual examples of studies using these experiments – especially if there are studies out there testing corals ‘in a variety of reef types’ as you state

>Thanks, we cited more short-term heat stress studies throughout the paper and removed the phrase “variety of reef types” here.

Line 544: ‘such delayed’ is confusing, should it just be ‘delayed’

>Thanks we made the change.

Line 561: While I agree with this statement, part of it feels a little repetitive of prior statements lines 506 and 527. Also, if you do keep this, maybe reduce the number of examples in each set of parentheses as this sentence is very long

>Thanks, we rephrased this sentence and the previous statements.

Line 575: Could change ‘of the stress reaction’ to ‘of coral stress phenotypes’? Just a suggestion

>Thanks, we like the suggestion and made the change.

Figures:

Figure 2: Is there a clearer way to present this? It isn’t the easiest figure to follow

>We appreciate the reviewer’s comment. We do think presenting the bleaching scores of heat stressed corals as a time series is important for representing one of our main takeaways—that bleaching state is stable following heat stress. To improve clarity, we changed the figure colors and altered the figure caption. 

Figure 5: Could you edit this figure so that the individual points are aligned with each violin? Also in the figure legend, line 415 – ‘with in between VBS scores’ reads awkwardly, would suggest rephrasing

>Thanks we incorporated both suggestions.

Supplement:

Figure S3: I think there’s an error with R2 and p-values as they’re the same in both panels

Thanks, the R2 and pvalues are confirmed to be the same. This is because no coral sample died between the Day 7 and Month 1 post-stress timepoints. We added this note to the figure to clear up confusion.

Reviewer #2: General Comments:

The authors present a comprehensive analysis of how immediate responses of corals to short-term (48hr) heat-stress correspond to responses over a week and one and two months post stress. The study is well designed, the data are clearly presented, and the conclusions are generally well-suited to the data. One nuance that may be worth spending a little more time discussing is the fact that the experimental design does not allow for examination of how "recovery" as examined in this experiment relates to in situ/ecological recovery from a natural bleaching event. Here, the data show clearly and convincingly that some corals are capable of recovering from a short-term stress exposure, which in and of itself is an interesting finding. However, without comparison data from a longer-term experiment or natural bleaching event it is difficult to draw conclusions on how the patterns of recovery herein might relate to natural recovery. To be fair, I don't think the authors have gone too far in interpreting their results, only that another sentence or two could be devoted to discussing how to build on these current results to better relate short-term recovery to ecological recovery (perhaps in the future directions section).

Aside from a few minor technical and grammatical comments noted below which should be easily dealt with in a minor revision, I support acceptance and publication of this manuscript.

Sincerely,

dan barshis

Line by line comments:

Introduction:

Line 64, species "composition" or "identity"?

>Thanks we made the change.

Lines 86-87, maybe change to "... stressed to a recovery state" as coral mindset seems a bit anthropomorphic

>Thanks we made the change.

Line 94, "In this study[,]"

>Thanks we made the change.

Methods:

I suggest considering use of the term "ramet" instead of "nubbin" as ramet additionally specifies that nubbins are from the same parent colony vs. nubbins could be from a mix of colonies. I've moved to using "colony" and "ramet" unless I know for sure that "colonies" are unique "genets" but it's up to you if you want to stick with the current terms.

>Thanks for the suggestion. Each “nubbin” mentioned comes from a unique colony. We initially determined unique genets through mitochondrial genome sequencing and later confirmed with SNP genotype calling in another study (in review but not uploaded to a public database, so we have now included the mitochondrial haplotype results and referenced genotyping in supplemental table 1). We have changed language to reflect unique genets.

I also highly recommend including a github repository with all of your R code so that people can properly recreate your analyses.

>Thanks for the note, we have now created a github repository with R code used for statistical analyses. It will be private until acceptance of the manuscript.

Line 168-169/182-183, do you have a citation for the identified symbiont cells divided by total cell counts methodology? I'm a little unclear on the details of the calculation. Wouldn't a better ratio be the number of symbiont cells to non-symbiont (i.e., host) cells? I can see how the symbiont/total cell ratio would be somewhat proportional to symbiont cell density per unit host tissue, however, with this ratio the relationship would not be linear correct? Consider the hypothetical scenario where you have two identically sized coral fragments (i.e., same amount of host tissue) and one has many symbionts and one is mostly bleached. In this case the bleached one would have a very low symbiont/total cell ratio while the densely populated one would not have as high a ratio as it should because the total cell number (denominator) is greatly increased by the number of symbiont cells. I might be wrong here but either a citation comparing this metric with more commonly used metrics or some additional acknowledgement of the limitations of this method would be useful. I think as a rough proxy for actual cell densities it should be fine but would be important to acknowledge the caveats so that other groups don't naively take it up as a direct replacement for a true symbiont density per host biomass measure.

>Thanks for the interesting points. Based on our knowledge the methodology we used is also considered a common symbiont concentration metric, though an important distinction is mentioned here between proportion as we used vs. ratio as was mentioned above. We added citations to corroborate our metric and changed language throughout the manuscript to not explicitly reference density.

Line 202, please include what kind of PAR sensor was used to measure the light levels (e.g., planar or spherical) as this affects the values recorded. Also, these light levels are quite low compared to average PAR values on the reef. How might this have affected your results?

> The PAR sensor is an Apogee Instruments Underwater Quantum Flux meter, model MQ-210 and planar. We have added this to the methods. We remeasured the tank system (previous values came from another study also conducted in the same tank setup—Cornwell et al., 2021) and found that the values were 53-94 μmol photons m-2*s-1. We have also added another caveat in the discussion addressing the lower light intensity measurement, although we don’t expect this value to have significantly impacted results over a 2 day heat stress experiment.

Lines 220-222, were light levels in the holding tanks ever recorded? Would be curious how they compared to your experimental tank light levels.

> Thanks for the question. We unfortunately did not measure light levels in the holding tanks. We relied on natural sunlight for holding tanks.

Lines 242-244, how was an ANOVA used to determine correlation, looks like it was the lme model maybe that generated the correlation in Table S2 not the ANOVA?

>Thanks for asking for clarification. We determined correlation using the lme model and then compared VBS groups’ symbiont concentrations using the ANOVA. We rephrased this sentence.

Line 244, please specify factors included in the lme model (i.e., what was included as fixed versus random effects)

>We state at the end of the statistical analyses section that all linear mixed effects models had reef region as a random intercept. No additional fixed effects are specified.

Line 261, suggest adding a sentence to the tune of "Specific model formulas and outputs can be found in Table S2."

>Thanks, we made the change.

Results:

Lines 264-279, suggest including the R2 here as well as other relevant results of the statistics in Table S2. Wording such as "highly distinct" or "little distinction" would be better clarified with p-values.

>Thanks, we incorporated the suggestion.

Lines 283-290, suggest specifying how this wording corresponds with your numerical categories here to avoid the reader having to flip back and forth to see how they align.

>Thanks, we incorporated the suggestion.

Line 314, please clarify whether this figure includes only heat treated or heat treated and control fragments.

>This figure only includes heat stressed fragments, i.e., no control fragments. We incorporated the change.

Line 328, I suggest including the 1 mo and 2 mo timepoints on here with a broken x-axis. The mortality data aren't really presented elsewhere in graphical form. You could also look into a cox proportional hazards/survival curve approach/figure instead as that is more common for this kind of mortality over time data.

>Thanks for the suggestion, we added the 1 mo and 2 mo timepoints to Fig. 3b.

Lines 353-355, I'm a bit confused on what data this last sentence is referring to. Above you discuss Day 0 data vs. Day 2/Day 7/Day 60 but I don't see non-Day 0 vs. Day 7 comparisons?

>Thanks, we removed this sentence for clarity.

Lines 360, "coincide[d]" no?

>Thanks, we made the change.

Lines 404, suggest changing to "... forty-nine out of fifty [surviving] heat stressed nubbins ..."

>Thanks, we made the change.

Lines 424-425, suggest adding "[reviewed in]" in front of the Grottoli reference.

>Thanks, we made the change to include more short-term heat stress experiment studies.

Lines 426-427, why 3 days here when above you focus on day 7?

>Thanks, we changed the sentence to day 7. We aimed to emphasize that most mortality happened within 3 days when looking at a 7 day up to 2 months post-stress period. We now emphasize this finding later in the discussion.

Line 431, what's the difference between recoverability and recovery? If none, suggest sticking with recovery to avoid introducing another term.

Thanks, we use them interchangeably so we will now stick with recovery.

Lines 455-457, could be varying starting pigmentation or it could just be the naked eye is unable to reliable distinguish between categories 1 vs. 2 and 4 vs. 5. Suggest adding wording to this effect.

>Thanks, we added the caveat.

Lines 471, I was a little confused when I first read this sentence. Suggest adding "improved bleaching categories [on day 7 vs. day 0]."

>Thanks, we made the change.

Lines 499-506, Either here or earlier on, I suggest referencing Evensen et al_2021_Limnology and Oceanography_Remarkably high and consistent tolerance of a Red Sea coral to acute and chronic thermal stress exposures as showing that both bleaching and sub-bleaching physiological responses can be consistent between very short and medium term stress exposures.

>Thanks very much for the citation. We have now referenced Evensen et al. 2021 in the introduction and discussion sections.

Lines 522-530, I'm missing some discussion of how your in-tank survival/recovery might relate to in situ recovery/survival. I think it would be best in this paragraph simply to acknowledge, remind the reader that your experiment looked at recovery in a controlled environment, and that recovery may have looked different if your fragments had been returned to the reef.

>Thanks for the comment, we have added another sentence to the paragraph to address in tank vs in situ recovery.

---

## [Editor Report · Decision Letter 1]

24 Aug 2022

Persistence of phenotypic responses to short-term heat stress in the tabletop coral Acropora hyacinthus

PONE-D-22-13834R1

Dear Dr. Walker,

We’re pleased to inform you that your manuscript has been judged scientifically suitable for publication and will be formally accepted for publication once it meets all outstanding technical requirements.

Kind regards,

Christian R. Voolstra, Ph.D.

Academic Editor

PLOS ONE
---

## [Editor Report · Acceptance letter]

1 Sep 2022

PONE-D-22-13834R1 

Persistence of phenotypic responses to short-term heat stress in the tabletop coral Acropora *hyacinthus*

Dear Dr. Walker:

I'm pleased to inform you that your manuscript has been deemed suitable for publication in PLOS ONE. Congratulations! Your manuscript is now with our production department. 

Kind regards, 

on behalf of

Dr. Christian R. Voolstra 

Academic Editor

PLOS ONE